# Exploring the challenges in the management of childhood pneumonia-qualitative findings from health care providers from two high prevalence states in India

Rani Mohanraj[1☉]*, Shuba Kumar[1☉], Monica Agarwal[2‡], Bhavna Dhingra[3‡], Saradha Suresh[1‡]

1 Samarth, Chennai, Tamilnadu, India, 2 King George Medical University, Lucknow, Uttar Pradesh, India, 3 All India Institute of Medical Sciences, Bhopal, Madhya Pradesh, India

☉ These authors contributed equally to this work.
‡ MA, BD and SS also contributed equally to this work
* ranimohanraj@samarthngo.org

**Data Availability Statement:** Request for data could be sent to Ms. K.V. Sri Priya, Administrative

## Abstract

India ranks among the top five countries in the world in child deaths due to pneumonia. Apart from poor public awareness, inadequate health infrastructure and treatment services have compromised effective management. This qualitative study guided by components of the Andersen-Newman's health care utilization framework explored contextual and community challenges faced by health care providers (HCPs) in the delivery of care services for children with pneumonia in select districts of Uttar Pradesh (UP) and Madhya Pradesh (MP). Semi structured interviews (SSIs) and focus groups discussions (FGDs) were carried out with a purposive sample of HCPs selected from three districts in each state. The HCPs included doctors and community health workers (CHWs). All SSIs and FGDs were audio-recorded, with consent, transcribed verbatim, entered into NVivo and analysed using thematic analysis. A total of 15 SSIs were conducted with doctors and eight FGDs were carried out with CHWs. Two themes that best explained the data were:, 1. Health systems: barriers faced in delivery of care services and 2. Evaluated Need: perceptions on community awareness and practices. According to the HCPs inadequacies in government health infrastructure both manpower and equipment, and skill deficits of paramedical staff and CHWs contributed to poor quality of care services for management of childhood pneumonia. This combined with inadequate understanding of pneumonia in the community, dependence on unqualified care providers and distrust of government hospitals acted as barriers to seeking appropriate medical care. Additionally, poor compliance with exclusive breast feeding practices, nutrition and hygiene had contributed to the high prevalence of the disease in these states. Strengthening public health facilities, instilling trust and confidence in people about the availability and the quality of these services and enhancing health literacy around childhood pneumonia would be critical towards protecting children from this disease.

officer, Samarth. She can be contacted at
admin@samarthngo.org.

**Funding:** This study was funded through a grant from the Bill and Melinda Gates Foundation (Grant No. OPP1084307) to the INCLEN Trust International with a sub-grant to Samarth (Subgrant No. INC2015GNT002). The funder had no role in the study design, data collection and analysis, decision to publish or preparation of the manuscript.

**Competing interests:** The authors have declared that no competing interests exist.

## Introduction

Every year, millions of children in the world less than five years of age die due to pneumonia. In 2018 more than half these deaths were reported from Nigeria, India, Pakistan, Democratic Republic of Congo (DRC) and Ethiopia. India recorded 127,000 deaths of children less than 5 years of age due to pneumonia during that year [1]. The Global Action Plan for Pneumonia and Diarrhoea (GAPPD) advocates the, 'protect, prevent and treat' model for management of pneumonia and diarrhoea with the goal of reducing pneumonia mortality from the current 5/1000 to 3/1000 live births by 2030 [2]. The model advocates protection through six months of exclusive breast feeding (EBF), adequate complementary feeding, and vitamin A supplementation; prevention through vaccination, reduced household pollution, provision of safe drinking water and sanitation and treatment through improved care seeking practices, referrals and appropriate case management at the facility and community levels. Studies from countries with a high burden of childhood pneumonia, including India have shown poor recognition of pneumonia symptoms and delayed care seeking on the part of mothers. Besides, these countries also have inadequate health infrastructure, sub-optimal treatment services, drug stock outs, poor provider motivation and high workload, all of which severely compromise effective management [3–5]. Many programmes initiated by the Government of India to reduce child mortality and morbidity since independence in 1947 have seen considerable success. The Integrated Management of Childhood Illnesses (IMCI) developed by WHO and UNICEF was adapted to include neonates and the Integrated Management of Neonates and Childhood Illnesses (IMNCI) was launched in a phased manner in India beginning 2003. The three main components of this programme are 1. Improving the skills of health care providers 2. Improving the health systems and 3. Promoting family and community health behaviours and practices [6]. However, despite the impetus and focus of the IMNCI programme to manage childhood illnesses including pneumonia, its delivery has not been optimal or uniform across all the 29 states of India. A case in point are, the states of Uttar Pradesh (UP) and Madhya Pradesh (MP) which have higher Infant Mortality Rate (IMR: 40 and 50/1000 live births respectively) than the national average(32/1000) [7]. Towards addressing this issue, the India Integrated Action Plan for Pneumonia and Diarrhoea (IIAPD) was developed along with WHO and UNICEF in 2014 particularly focussing on the four states of Uttar Pradesh (UP), Madhya Pradesh (MP), Bihar and Rajasthan, which account for more than half the deaths of children less than 5 years of age in the country. A Lancet communication stated that in the year 2015, more than half the children younger than 5 years had pneumonia (565 cases per 1000 children [95% Uncertainty Interval 94–2047] in Uttar Pradesh; 563 cases per 1000 children [UI 88–2084] in Madhya Pradesh) suggesting a compelling need to focus on the management of this disease in these two states [8].

This paper is part of a large mixed methods study that included quantitative surveys and qualitative interviews. It was carried out in select districts in three Indian states UP, MP and Tamil Nadu (TN), with a sample of mothers with children less than 5 years of age with probable pneumonia to document their care seeking behaviours. The data collection was carried out between April 2016 to January 2017. Findings of the quantitative survey revealed better care seeking behaviours and government health service utilization in the southern state of TN whose IMR is better (19/1000 live births) as compared to UP and MP. The majority of mothers in UP went to untrained care providers (UCPs) while those in MP preferred private allopathic care. We were however, unable to ascertain if these private allopathic care providers were qualified medical doctors or not [9]. Difficulties in accessing government care facilities, non- availability of doctors and medicines and long waiting time were some of the reasons reported by mothers in qualitative interviews, for poor utilization of government care facilities [10]. Given

the enormous responsibility borne by the public health systems in these two states to bring down both the incidence of childhood pneumonia and the IMR, this paper, examined the perceptions of government health care providers on challenges they faced in the effective delivery of pneumonia care and their perceptions on community practices that exacerbated the risk for childhood pneumonia.

## Conceptual framework

The Andersen-Newman framework of health care utilization guided the conduct of this study. According to this framework, health care access and utilization can be assessed based on three characteristics, namely predisposing factors, enabling factors and perceived and evaluated need [11]. The components of predisposing, enabling and perceived need were explored with the mothers of children who were less than 5 years of age and have been described elsewhere [10]. This paper which explored the perceptions of health care providers was guided by the health systems and the evaluated need component of the framework which refers to how health care professionals judge people's health status and need for medical care. The framework also incorporates the health service system, its health care personnel, equipment and materials and how these are used in providing health services (Fig 1). For the purpose of this study we specifically explored the challenges faced by health service personnel working in government hospitals in the selected rural areas of UP and MP in the delivery of pneumonia care for children less than 5 years of age.

## Materials and methods

### Ethics statement

Ethical approvals were obtained from the respective ethics committees of the study sites, namely KGMU institutional ethics committee (74th ECM II-A/P13), King George Medical University, Lucknow, Uttar Pradesh, Institutional human ethics committee, AIIMS, Bhopal, Madhya Pradesh and Institutional Ethics Committee of Samarth, Chennai, Tamilnadu. Most

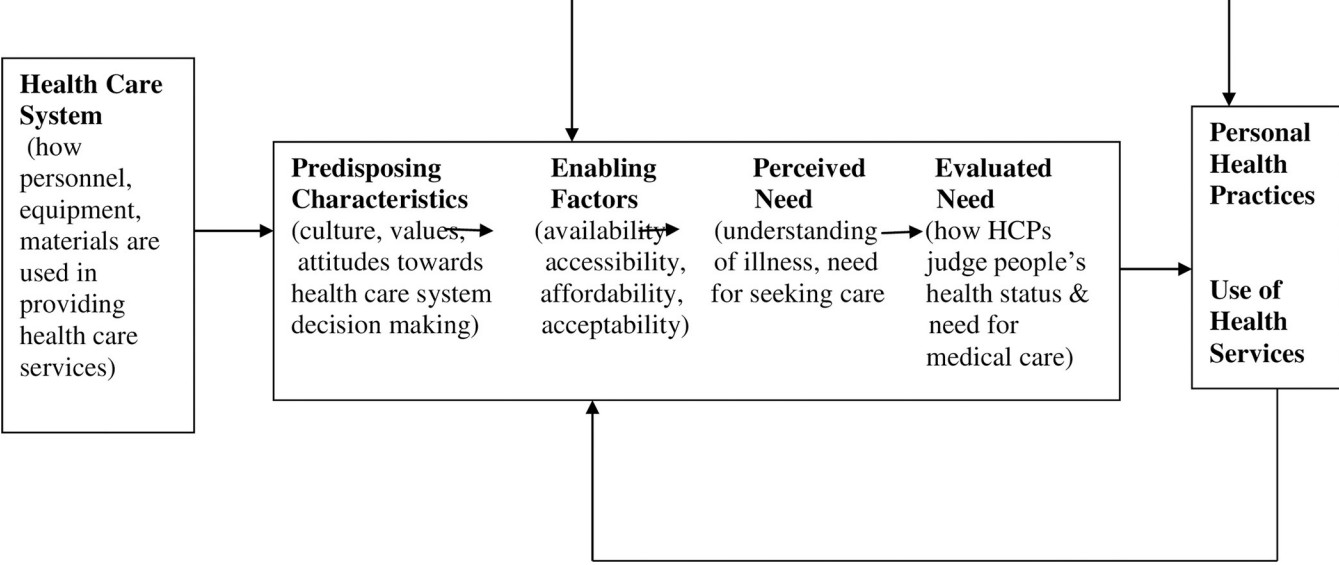

**Fig 1. Conceptual framework- Andersen and Newman framework for health service utilization.**

of the doctors provided consent to audio record. Where permission to audio record was denied, notes were taken during the interview.

## The context: Rural public health system

Rural areas have a three tier system, namely the i) health sub centre (HSC), ii) the primary health centre (PHC) and the iii) community health centre (CHC). The HSC is the most peripheral contact point between the primary health care system and the community. It serves a population of around 5000 and is staffed with one female Auxiliary Nurse Midwife (ANM) and one male health worker. These health sub-centres are provided with basic drugs to address minor ailments and are tasked with creating health awareness, guiding and encouraging better health services utilization in the community. The PHC serves a population of about 30,000 (20,000 in hilly and difficult to reach areas) is staffed with one medical officer (MO) who is supported by about 14 paramedical and administrative staff and acts as a referral unit for 5–6 HSCs. It has 4–6 beds for patients. Activities involve health care promotion and curative services. Lastly, the CHCs cater to a population of around 120,000 (80,000 in hilly/difficult to reach areas) are staffed by four medical specialists, namely, physician, gynaecologist, paediatrician and a general surgeon. They are supported by 21 paramedical and other staff, are equipped with 30 beds, have an operation theatre, labour room and x-ray and laboratory facilities. They serve as the referral centre for PHCs within the block. Lastly, every district in India has a district hospital, which provides comprehensive secondary health care, is staffed with specialists and serves as a referral hospital for the district [12]. The health system in India is significantly dependent on the community health workers. The ANM is a multipurpose female health worker who is the link between the health system and the community. The ANMs are placed at HSCs (serve a population of around 5000) as well as at PHCs (serve a population of around 30,000) and are required to live in the vicinity of the HSC enabling them to be available for the community round the clock. Their responsibilities include maternal and child health care, immunization, school health, education and counselling. The Accredited Social Health Activist (ASHA) is also a CHW who is usually a woman who works to improve health status of members in her community. They are trained on childhood illnesses and are expected to identify conditions like pneumonia and diarrhoea in children, treat non severe cases and refer severe cases to the hospital for management [13].

## Study sites

Three districts of UP (Kanpurnagar, Faizabad, Shravasti) and three of MP (Bhopal, Satna and Panna) were the study sites selected in consultation with the respective government health representatives. The selection of the HCPs was linked to the larger study sampling strategy wherein the HSC constituted the sampling unit. A list of all the HSCs functioning in the selected districts was obtained and 30 HSCs per district were selected using the population proportionate to size method for the purpose of the household survey. The government HCPs were selected from among the district level hospitals and CHCs functioning in these selected sites to which these HSC were attached and included both the MOs and the CHWs [10].

## Participants

Our sample of MOs comprised of paediatricians (specialists) and MOs posted in district hospitals and in the CHCs. They were approached to participate in semi-structured interviews (SSIs) which were conducted in their places of work. The CHWs included ANMs and the ASHA workers attached to the CHCs. They were approached for participation in focus group discussions (FGDs) which were conducted at the premises of the CHCs.

## Study instruments

Separate SSI and FGD guides were developed for use with the MOs and CHWs and were broadly based on health systems and evaluated need component of the Andersen and Newman framework. Thus, the guides explored their perceptions about reasons for the high burden of childhood pneumonia in their districts, challenges and gaps they faced in its management and treatment which influenced delivery of care services. Their perceptions and awareness about signs and symptoms of pneumonia in the community, its influence on care seeking practices and their thoughts on the practices of breast feeding, hygiene, nutrition and immunization were also elicited. Consent to participate and to audio record the SSIs and FGDs was obtained from each participant prior to the session. The SSIs and FGDs were conducted by the respective site investigators based at King George Medical University (KGMU), Lucknow, UP and the All India Institute of Medical Sciences (AIIMS), Bhopal, MP, each of whom had been trained in qualitative interviewing skills by RM and SK. All interviews with the MOs were carried out either in English or the local language (Hindi) and all FGDs with CHWs were carried out in the local language Hindi. All the audio recorded interviews were transcribed verbatim, translated into English and analyzed using NVivo software (version 8) by RM and SK.

## Analysis

Using a hybrid deductive-inductive thematic analysis strategy we first created a code book based on the health systems and evaluated need component of the Andersen -Newman Framework. Following data collection we applied the thematic analysis approach developed by Braun and Clarke [14], which included the seven steps of transcription, reading and familiarization, coding, searching for themes, reviewing themes, defining and naming themes and finalizing the analysis. Both RM and SK read through each transcript to gain familiarity with the context and issues described therein. They then inductively coded three transcripts independently with each developing a coding scheme, expanded the existing code book and added codes and categories inductively derived from the interviews. These were reviewed and discussed and a single code book to be used for coding all subsequent interviews was finalised. New codes were included when new issues emerged while coding the remaining transcripts. Once coding of the entire dataset was completed we began to search for patterns and themes in the data which would help to answer our research objectives. Once the themes were identified we further refined the process of theme identification by defining the nature of each theme & their relationships with each other. Suitable quotes were extracted that helped explain and illustrate these themes.

## Results

A total of 15 SSIs with doctors (7 in UP and 8 in MP) and eight FGDs with CHWs were carried out. Wherever possible we conducted separate FGDs with ASHAs and ANMs. Where their numbers were less, we combined ANMs and ASHAs in one FGD. Each FGD had an average of about 8–12 participants (total 82 participants). In UP, we carried out 5 FGDs (3 with ASHAs; n = 32 and 2 with ANMs: n = 14) and in MP we conducted 3 FGDs (mixed group of ASHA and ANM: n = 36).

### Demographic characteristics of medical officers

**Uttar Pradesh.** A total of 7 MOs, all males, from the study districts participated in the SSIs. They included one paediatrician, each from the district hospitals of Kanpurnagar, Faizabad and Shravasti and four MOs working in CHCs. Their ages ranged from 28 to 52 years

(median age = 38). Two paediatricians had completed their post graduate diploma in child health (DCH) while one of them had a post graduate degree in medicine (MD). The MOs in the CHC had completed their undergraduate degree (MBBS) while the MO in Faizabad had a diploma in Child Health (DCH). In terms of their years of experience in the current position most had completed five or more years barring the paediatrician from Shravasti district hospital who had completed 1.5 years.

**Madhya Pradesh.** A total of 8 MOs, all males from the study districts participated in the SSIs. They included one Chief Medical Health Officer (CMHO) each from the district hospitals of Bhopal, Panna and Satna as well as one District Health Officer (DHO) from each of these districts. In Panna the Block Medical Officer (BMO) and the MO from the CHC also participated. Their ages ranged from 30–55 years. The ages of two HCPs were not recorded. Four HCPs had an undergraduate degree while the other four had a post graduate degree (MD/DCH). The experience of these HCPs in their current position ranged from 1–8 years

### Demographic characteristics of ASHA and ANM

In UP, the minimum age of the CHWs was 25 years (Faizabad) and maximum was 58 years (Shravasti). Their Education ranged from a minimum of 8 years of schooling to a maximum of a master's degree. In MP, the minimum age was 26 years (Bhopal) and the maximum was 59 years (Panna). Education here too ranged from those who had only 5 years of schooling to those who had a post-graduate degree.

### Themes

Using the evaluated need and the health systems components of the Andersen and Newman Framework for health service utilization we present our findings following analysis of our interviews with HCPs under two thematic heads:

1. Health Systems: Barriers faced in delivery of care services

2. Evaluated Need: Perceptions on community awareness and practices

   1. Health Systems: Barriers faced in delivery of care services
   These are described under the following two sub-themes, i) systemic barriers ii) skills training and supervision
   i. Systemic barriers: These have been explained under two specific heads of a) equipment and drugs and b) human resources
   a) Equipment's and drugs:
   Shortages of drugs (antibiotics), and health care personnel (staff nurses, support staff and doctors), inadequate availability of oxygen cylinders (in district hospital) and absence of separate wards for children were some shortcomings reported by almost all the MOs in the CHCs. The MO in the Kanpur Nagar CHC, said, "*Fourth line antibiotics are available but in less numbers. . . .. If we want to use amoxicillin or penicillin tablets they are not available, injections like amikacin or gentamycin are available but in limited quantities*". Further, as described by an MO at the district hospital in Shravasti (UP)- a high prevalence district- the non-availability of some prescription medications at the hospital required patients to purchase them from private pharmacies leading patients to complain, *'if I have to buy it from outside why should I come here'*. The absence of a 24 hour pathology laboratory and X-Ray facility were other handicaps at the district hospital in Shravasti (UP) that affected delivery of appropriate care. A few MOs stressed the need for radiological facilities that could enhance their ability to confirm diagnosis of severe pneumonia. In MP, the MOs in Panna district reported absence of X-ray machines, low stocks of antibiotics and shortage of skilled man power in their CHCs. According to the

CMHO only two CHCs, out of six in Panna district had X-ray facilities during this study period. The MOs in Bhopal and Satna however, expressed satisfaction with the facilities available for management of childhood pneumonia in their districts. The CMHO of Satna attributed this to the effects of the National Health Mission programme which *"has made available good facilities in our hospital at community level, also at district level and we are doing good work in this situation"*.

The most common barrier reported by the CHWs from both MP and UP was the poor and often erratic supply of medicines in their PHCs and CHCs. They believed that absence of a regular supply of Cotrimoxizole recommended for children with respiratory problems, had affected their ability to deliver proper care. The ANMs said that availability of antibiotics both with them and at government health facilities would help build the community's faith and trust in government facilities thereby enhancing appropriate care seeking by the community. Quite often they were forced into referring mothers to government hospitals located further away due to non-availability of drugs. As CHWs in an FGD from Kanpurnagar (UP) reported, *"R3: We don't have any medicines with us. . .If we have medicines with us, then everyone will come to us. . .R 4: Except paracetamol, nothing is given, that too is not given sufficiently. R5: In the last 6 months we never received amoxicillin tablets. Last year we received syrup once. Small bottles of 10ml are what we get, how is it adequate? R6 When the expiry date of the medicine is near..we get medicines that could expire like 2–3 months"*

Non availability of nebulizers and oxygen therapy required that mothers' had to take their child to a higher facility which involved transportation and added costs. People therefore tended to prefer going to private doctors who were both accessible and available, *"instead of suffering in the government"* a sentiment echoed by many CHWs in Faizabad. In Satna (MP) one MO stated that the problem of accessibility to government health care facilities was a major barrier for delayed care seeking. Despite the presence of the 108 ambulance facility, there had been cases when the ambulance had not reached on time to pick up the child either because it had been held up elsewhere or the drivers had delayed reaching the place for personal reasons. The MO stressed the need for establishment of more government health facilities and for increasing the numbers of ambulances under the 108 ambulance scheme.

b) Human resources

Inadequate numbers of staff were reported as contributing to the heavy work load of the existing personnel which in turn resulted in sub-optimal care delivery especially during the monsoon and winter seasons when the numbers of children presenting with respiratory problems was high. In this context the CMHO of Satna district (MP) said, *"Patients are more and manpower is less. We are working at 40% manpower whether it is doctor or staff nurse. So our care is little affected. We have 50 patients and 2 staff nurses then care delivery is a little weak. We have expectation from the system because as per the WHO, the ratio of beds to doctors should be there. If this is fulfilled then we can do good work."* Shortage of nurses compromised proper monitoring of patients resulting in a hesitancy to admit children in Kanpurnagar CHC. As stated by an MO in this facility, *"There is no monitoring system here, we don't have any staff nurse with us who can monitor, there is no staff separately who can be in ward and give injections timely and inform me on time. . . only 1 sister remains here 24 hours, here they are only for delivery. Most of the time they are so busy that they are not even able to give injectiosn on time. That is why we are afraid of admitting patients here and especially children who need to be monitored carefully"*. The issue of not employing permanent health workers was perceived as another impediment to providing care and maintaining hygiene in government health facilities in these states. Nurses, sanitary workers, attendants etc were posted on a contract basis. At the end of the contract period, quite often, new workers tended to get appointed with limited exposure to formalised training which affected both the maintenance of appropriate standards and hygiene.

While the MOs lamented on the inadequacy of nursing and other support staff, many CHWs in UP reported that mothers often complained of non-availability of doctors at the facility at all times, which caused them to seek care from local UCPs. The delayed attention given to patients was another concern echoed by the CHWs, "*R1:Here* [refers to government facility] *the system moves very slowly; the system comes first, treatment comes afterwards. R2:If some serious case comes, first we have to complete the document whereas in private though they take money, treatment starts immediately*". The ANMs of Faizabad (UP) and Panna (MP) spoke of the absence of a 24/7 government health facility and of a paediatrician at the CHC. Repeated experiences of being turned away or of being referred to higher facilities were other reasons why the community preferred not to seek care at government facilities.

ii. Skills training and supervision

Most doctors from both the states highlighted the importance of consistent training for effective delivery of care for childhood pneumonia which was not carried out. While the scope of our study was not to assess doctors' knowledge about diagnosis and treatment of pneumonia, all doctors predominantly relied on the presenting clinical symptoms to arrive at a diagnosis. A majority of the doctors from the three districts in UP and many from MP reported not attending any specific training programmes on childhood illnesses including pneumonia: "*I have not got any training in the last five years. IMNCI* [Integrated management of neonatal childhood illnesses] *has not happened. I think I attended the training in Jabalpur around 7 years back but don't have any certificate*" (Block Medical Officer, Panna, MP

The DHO from Satna (MP) while stating that training for the different categories of providers had been carried out, felt the need for refresher training programmes to strengthen skills and enhance the quality of care, when he said, "*Till now, training has been given to Aanganwadi workers, ASHAs. But even after training they do not have the skills to identify Pneumonia properly. At least they should be able to give treatment there and send serious cases to the hospital. We have given them the tablets-cotrimoxazole..but if they are not able to identify, how would they give the medicine correctly to the children*? *They do not have this knowledge though training has been given*". Another challenge in care delivery reported by them was the low level of competence and skills among pharmacists and nurses which seriously compromised quality of care. According to a doctor in a district hospital in Shravasti (UP) many nurses did not have the right educational qualifications. Some did not even have a science background. Poor knowledge of spoken and written English made it difficult for them to follow the doctor's orders which were usually written in English. Many MOs stressed the importance of field level supervision and monitoring for CHWs. The MOs from both states were of the opinion that poorly trained CHWs in the community resulted in poor identification of childhood illnesses and referrals to health facilities contributing to delays in appropriate care seeking and to people's dependence on the UCPs. At the health facility level, the government doctors requested the need for capacity building of nurses and technicians as many of them lacked basic skills like administering injections to children. They also emphasized the importance of supervision of nurses and lab technicians. One MO in a CHC in Faizabad shared his frustration when he said, "*The pharmacist who is here. . . many times don't even know the appropriate antibiotic. . . what is to be given to the child of that age-group they don't even know that. Staff does not know how to do nebulization, how much is to be given, how much ml is to be taken, how much "sorvitamol" is to be taken etc. Most probably they won't be able to do it.. ..Yes, we will have difficulties in the in-patient ward too. . . we don't have it* [nebulizer] *it and we need it. We don't have expert staff who can monitor everything, who can identify all symptoms, that "pasalichalrahi"* [fast breathing] *is there or not;""naakbehrahi"* [running nose] *is there or not. . . the one who can differentiate between all those things . . .. such staff is not there.*"

The CHWs from both states in turn indicated that they had not attended any training programmes. Excepting a few ASHAs and a couple of ANMs in a few localities in both states, most spoke at length of how important it was for them to be updated and provided training on various aspects of childhood illnesses. While they had received training on management of diarrhoea, relevance of exclusive breast feeding practices and hygiene, no specific training on pneumonia management had been held. Acknowledging the need for such training and their lack of preparedness CHWs in Kanpurnagar (UP) said, "*Our training should be done. We should be able to identify pneumonia early, not when they get serious, if we identify the disease early then it will be beneficial for us and related to that if we get the medicines to administer to the child at home, it will be really good. . .delay can be averted. . .. so our training is necessary, when we don't have knowledge then what is the use of us in the field*". Although a few of them described being informed about fast breathing and congestion as symptoms indicative of pneumonia most CHWs were not confident about identifying these symptoms. They felt it was imperative that they be provided more updates on childhood pneumonia for early symptom identification, care provision and referral. Some concerns the ANMs reported which they believed impacted delivery of their services was the poor living conditions made available to them in the vicinity of the HSC. Often these accommodations lacked proper sanitation, was in disrepair and had poor internet facilities. The ANMs were therefore reluctant to stay here and instead found alternative accommodation elsewhere, thereby affecting their easy access to the communities they were meant to serve. Further each ANM was responsible for many areas but they could at the most visit an area about twice in a month and were heavily dependent on the ASHAs for information about the health of children in these areas. The ANMs in Panna and Satna pointed out that the terrain was hilly and vast with poor public transportation services which made it difficult for them to reach out to the families living in these far flung areas. Recruitment of more CHWs was strongly reiterated.

2. Evaluated Need: Community awareness and practices

These are described under three sub-themes, i) perceptions on community awareness and understanding of childhood pneumonia ii) perceptions on exclusive breast feeding and immunization practices in the community iii) perception on nutrition and hygiene practices in the community

i) Perceptions on community awareness and understanding of childhood pneumonia:

Poor awareness about the symptoms of pneumonia and its severity among mothers, delays in seeking appropriate care, trust in UCPs and non–adherence to the dosage prescribed by allopathic doctors were challenges reported by MOs and CHWs alike. The DHO of Satna (MP) spoke about the practice of relying on home remedies and indigenous treatments in remote area and said, "*There is a lack of awareness about pneumonia, specifically in remote areas plus they follow indigenous treatments like using 'sambhar kesingh*" [medicine prepared from horns of deer]. *Some give haldi* [turmeric], *some rub garlic etc. they do all such things. When the child is serious they give brandy to drink,. . .all this is done. When condition is critical then they take to the hospital*"

The predominant observation was that quite often families did not follow the recommended dosage and expected miraculous results. According to CHWs from both the states, the poor economic situation of families was the reason for poor adherence to prescribed treatment and to seeking care from UCPs who were available, affordable and accessible. People accepted and trusted them. They were seen by many as the first point of care in the community. The UCPs often tended to prescribe antibiotics which sometimes led to complications, further compromising the health of the child. In MP, the CMHO of Bhopal stated that many mothers took their children to so called 'private medical practitioners' many of whom were unqualified and dispensed allopathic medicines, resulting in more harm than good for the child. A paediatrician at the district hospital, of Shravasti (UP) lamented," *People come in late*

*and I am not in a condition to do anything at that time. Like when the child comes in gasping stage or sudden respiratory distress with only* [count of], *2–4 heartbeats present. They first consult the quack* [UCP]*when there is no relief from quack then they try other people nearby. One more thing I have seen that people in the community give suggestion-lets go and consult so and so then child will be all right. . ..people roam here and there and finally come to district hospital. There are many families that do not have a means of transport of their own. But now due to 108* [ambulance service] *they come. But 108 too come late sometimes. Families say that the vehicle* [108] *came late or sometimes they tell that it took a long time to arrange for a vehicle.*"

Many families in rural areas continued to provide home remedies and /or purchased over the counter medicines from local pharmacies which delayed delivery of appropriate care and/ or worsened the illness in the child. According to the HCPs low literacy in the community led to poor understanding of the instructions on dosage and the importance of completing the course of medication as prescribed. However, some CHWs in Bhopal (MP) stated that a large number of mothers' consulted them first and took their children to a doctor even when the child had a mild temperature.

ii). Perceptions on exclusive breast feeding and immunization practices in the community
These are described under the following two sub-themes, a) exclusive breast feeding and immunization, b) nutrition and hygiene

a). *Exclusive breast feeding (EBF) and Immunization*
The practice of six months of exclusive breast feeding in both states was not considered to be optimal. While MOs from Kanpurnagar (UP) said it was as low as 15–20%, the CHWs said the highest could be 75%. The majority stated that mothers' introduced top feeds or substituted with cow/buffalo milk well before the six months were up, sometimes even as early as the second or third month. The MOs highlighted the role of elder family members, especially the mothers-in-law, who continued to play a major role in deciding on feeding practices which impacted on exclusive breast feeding. The general belief in the community was that breast milk alone was inadequate for the baby after the 3rd or 4th month and top feeds needed to be introduced in the interest of the child's good health. Cultural beliefs about providing the baby a variety of home-made preparations like *Ghutt*i [ayurvedic herbs mixed with milk] etc were widely practiced and strongly believed to be good for the child. Added to this were other factors such as the poor nutritional intake of mothers leading to inadequate breast milk, necessity to switch to alternate milk because of having to work etc which contributed to early discontinuation of breast feeding. The BMO of Panna (MP) reported that among the tribal populations it was believed that '*colostrum causes illness in the newborn by entering the brain of the child*' which acted as a strong deterrent to giving it to the newborn baby, "*In institutional deliveries we can do it but in home deliveries we are not able to ensure that. Even in hospitals we have to insist a lot on giving colostrum. They do not give it..they discard it*".

With regard to immunization the HCPs were of the opinion that in recent years families were more proactive in completing vaccinations for their children. "*R1*: *Immunization is done regularly because many come to hospital for delivery. They never used to come for immunization few years back. . .. now they understand. R2*: *Communication about immunization has increased and so now awareness has increased*"(FGD, Satna, MP). A few CHWs said that sometimes families demanded to know why so many 'injections' had to be given and were concerned about the fever and rash that often followed these vaccinations.

One of the CHC doctors from Faizabad (UP) stated that while the pneumococcal vaccine (PCV) was now available it was not yet part of the universal immunization programme and being expensive was out of the reach of many families. The PC vaccine had been included in the Universal Immunization Programme in 2017 in few states in India which included Madhya Pradesh and only in select districts of UP.

b). *Nutrition and hygiene*

The doctors from both states believed that the poor economic situation of many families resulting from their low daily wages, unemployment, large family size and fewer opportunities for earning seriously compromised the nutrition of the child. A few of them were of the opinion that poor feeding practices prevalent in families also contributed to malnutrition. Many mothers were unaware of the nutritive content of breast milk and of locally available low cost produce. Although CHWs did talk about issues concerning nutrition of the child with mothers in the community, the need to include nutritional counselling in a systematic manner with the use of multimedia was believed necessary to strengthen awareness. As emphasised by the MO in Kanpurnagar CHC (UP), "*Most of the children who come are malnourished. I have to give 10–15 minutes to 1 patient for counselling . . . explain it to the mother. Most of the time faulty feeding is the cause if there is no other pathology . . . major reason is pneumonia here. . .You know there is a long line outside we are not able to explain to her nicely, along with this I have to see adult patient also.. I have to see general OPD also*"

Hygiene was also reported to be very poor in the community by HCPs from both states with very few practising hand washing, "*Hygiene is a major issue in newborns as the maximum mortality is in children below 1 month. In our newborn care, 50%* [of mortality] *is because of Sepsis*" (CMHO, Panna, MP). Low literacy levels and poverty compromised their understanding of the relationship between poor hygiene practices and diseases. The use of cotton dipped in cow's milk to feed the baby, use of dirty cloth to wrap or wipe the child with, were some harmful practices which increased the risk of infection in babies. These cultural practices were challenging issues which the HCPs struggled with on a daily basis and which contributed to increased mortality in children. A few MOs in UP agreed that hygiene was indeed an important issue but felt constrained by their clinical responsibilities which afforded them very little time to educate patients. The CHWs said that many families did not wash their hands with soap and usually only rinsed hands with water. The HCPs accepted the fact that affordability to buy soaps could be an issue for many in the community; nevertheless, developing awareness on hygiene practices and its relationship to illnesses and diseases in children was considered both important and necessary

A few MOs believed that, understanding about these protective and preventive measures was low even among ANMs, many of whom were not aware that their practice could contribute to a reduction in childhood pneumonia. Many ASHA workers were unaware about the different vaccines and the diseases for which they were intended. One of them from Panna (MP) said that, "*the child who is vaccinated with BCG has less chance of getting pneumonia*". In Satna (MP), both the district health officer DHO and CMHO described industrial pollution also as a major cause for pneumonia. In Shravasti (UP) and in Panna (MP), the MOs said that poor economic conditions in rural areas resulted in use of "*chula* [cooking on coal fire] *by 50% of the families*" a known risk factor for pneumonia. The paediatrician in the district hospital of Shravasti (UP) felt strongly about the association of poverty with pneumonia when he said, "*Yes, people are very poor here; this is one of the reasons for this problem. I think if poverty is not controlled the problem here will remain the same*". One of the MOs was of the opinion that improving economic conditions of the community would improve literacy levels and better health outcomes in the community.

## Discussion

Our study identified specific challenges faced by HCPs in select districts of UP and MP in the delivery of pneumonia care services suggesting that immediate attention is warranted by the

governments' in these states if the Sustainable Developmental Goals specific to childhood pneumonia are to be met. The key findings from the study can be summarised as follows:

- Poor government health infrastructure in terms of manpower, drugs, and equipment in some districts compromised quality and delivery of care services and its uptake by the community

- Skill deficits and inadequate training of paramedical staff and CHWs affected their ability to fulfil their roles as frontline workers

- The community's poor understanding of danger signs for pneumonia, their dependence on UCPs combined with their disillusionment with the government health sector have acted as barriers to seeking affordable and appropriate medical care

- Sub-optimal performance of practices under the 'prevent and protect model' with respect to EBF and poor nutrition and hygiene have contributed to high prevalence of the disease in these states

We believe that the above key findings highlighting both the inadequacy of resources as well as poor uptake of government health care services in the community, have relevance to other low and middle income countries (LMICs). Studies have shown that about 40% of very poor children with suspected pneumonia are not taken to a health facility in LMICs [15]. Care seeking is incumbent on early recognition of symptoms, decision to seek care from appropriate providers, accessibility, availability and affordability factors [16]. The absence of all or any of the above could result in delays in care seeking-more so among the economically deprived. Our quantitative and qualitative findings with mothers found that UCPs were the preferred choice among a majority of mothers who chose them for reasons of affordability, ease of access and previous negative experiences of poor services at the government facilities [9, 10] findings, which are similar to earlier studies in UP [17, 18] and other LMICs [19]. While majority of the doctors in our study felt that mothers had to be made aware to seek care from appropriate providers at government health facilities, their own admissions of poor health facilities impacting appropriate care seeking is a point of concern which is in line with a study conducted by Colaco and Mishra in UP and Bihar [20].

According to the Indian Public Health Standards, a CHC should provide 'routine and emergency care for sick children'. The minimum human resources that this facility is meant to be equipped with are, five specialists (including one paediatrician), two allopathic medical officers (MOs), ten staff nurses, one pharmacist, two lab technicians and one X-ray technician [21]. For the most part, such specialists and trained staff were not available in the facilities included in our study sites as reported by the MOs. A comprehensive situation analysis in select districts of five states, including UP and MP on challenges with respect to pneumonia care in government health facilities showed problems with respect to supply of drugs, lack of specific guidelines for CHWs in the use of antibiotics and inadequate number of specialists and staff at the primary and secondary health facilities [22]. Such a lack of services, manpower and equipment is suggestive of poor accountability and commitment from the respective governments which severely affected quality of services. It is important to highlight that the under-utilization of government facilities in these two states is also indicative of a deeper malaise in the community, one of distrust of the government health sector and of the fact that the community lacks the purchasing power to demand quality health services from the government health system in these states. Communities thus need to be made aware of their right to proper health care, and be empowered through advocacy and health education as these can lead to demand for quality health services in government facilities [23].

The doctors believed that harnessing the resources of the CHWs was vital and could be an effective strategy for reducing severe cases. The role of CHWs, both ASHAs and ANMs in new-born care and in reducing childhood illnesses (including pneumonia) is crucial as they have been tasked with improving community care practices through health education, early identification of illnesses and danger signs, administering pre-referral drugs and provision of appropriate referrals thereby preventing delays. Many challenges with respect to leveraging the resources of CHWs were reported. The doctors pointed out that the CHWs displayed poor knowledge and understanding of childhood pneumonia related to their roles and responsibilities. This combined with insufficient training and supervision contributed to their inability to fully execute their responsibilities. Training in IMNCI strategy and refresher training were grossly missing or was not regular for the ANMs. Further, they had not received any training on new guidelines. While both categories of CHWs had undergone training on a few management skills for childhood illnesses, many who joined subsequently reported no training or refresher programmes on pneumonia and other childhood illnesses, resulting in a lack of confidence among the CHWs, findings that have been reported in other studies as well [18, 20, 24]. Systematic reviews of community case management [25, 26] and studies from India [27, 28] have shown that training of CHWs can result in improved trust and better recognition of their role in the community. It contributed to a reduction in care seeking from UCPs and to a reduction in the severity of pneumonia due to early and appropriate care seeking. The Government of India guidelines for management of childhood pneumonia by ANMs involves treatment with oral Amoxicillin or Cotrimoxazole for five days for non-severe pneumonia and administration of first dose of amoxicillin followed by immediate referral to hospital for severe pneumonia. However, the ANMs in our study were hampered by inadequate supply of these drugs. They also had not received any training on these new guidelines. Equipping the ASHA workers with training in Integrated Community Case Management (ICCM) and providing medical kits with Amoxicillin and Oral rehydration tablets, improved their ability to identify and appropriately manage pneumonia [28]. However, if these training programmes are not sustained these effects do not last long [29]. One of the findings following an evaluation of the implementation of the IMNCI programme in several states in India was that poor supervision and monitoring were bottlenecks for implementation of the programme. Peer supervision and supervision by trainers of IMNCI modules are two innovative approaches that had been tried in few districts [30]. One other approach recently tried out is the e-IMNCI, an e-learning tool to improve clinical practices of ANMs and MOs in select districts in Jharkand, a northern state in India. Results of this pilot study has shown encouraging results in the use of the digital platform by ANMs in terms of improved knowledge in identifying clinical signs and symptoms, tracking illness in the community, maintaining adequate stocks of medicines, and providing appropriate referrals [31].

With respect to six months EBF as a potential protective factor and its sub-optimal practice, studies have shown low rates because of nutritional deficits in mothers [32]. Programmes that combine education and counselling interventions in communities and health facilities during ANC visits, promotion of baby-friendly hospitals, use of religious leaders in spreading messages can improve EBF rates [33, 34]. Earlier studies have shown an association between caregiver's personal hygiene practices with childhood malnutrition and diseases [35, 36]. Many doctors felt that the low literacy levels of mothers made understanding of health messages on hygiene, a challenge. The importance of relevant and simply articulated messages that would educate people and in the process get them to seek appropriate care has been proven to work in UP [27].

## Strengths and limitations

This study used the Andersen and Newman framework specifically focusing on HCP perceptions about the health systems within which they functioned and the evaluated need component which explored their perceptions on challenges they faced from the community which compromised care delivery. Using this framework gave us both the logic and the structure and enabled good insights into delivery of care services in these government settings in UP and MP. Using a hybrid approach of qualitative thematic analysis involving a combination of deductive and inductive strategies contributed to the rigour of our analytical approach. In terms of limitations our study findings can only be transferable to mid level health facilities like the CHCs and tertiary facilities like the district hospitals in rural settings in India. Including MOs working in PHCs which are the first level health services in rural areas, could have provided added insights about the challenges they face. An inclusion of a larger sample of health care providers with due consideration to an urban component could have contributed to a deeper understanding of the rural-urban health care scenario, perhaps helping to highlight unique features in each setting from which lessons could be drawn.

## Conclusion

With childhood pneumonia continuing to be a major contributor to infant mortality, particularly in some northern states, the time for reflections is past. It is the time for serious actions; actions aimed at strengthening the government health care infrastructure and instilling confidence in the minds of the people about both the availability and the quality of these services. Strengthening community based care and management through health workers as the principal delivery platform in rural areas would make appropriate health care accessible to many more in the community. These actions are very much in the realms of possibility as has been evident to a large extent in Tamilnadu where utilization of government health care services-though on the low side- is still far higher than in states like UP and MP. Combine this with good health literacy and we have an effective strategy that will protect our children from this preventable disease.

## Supporting information

**S1 Text.**
(DOC)

**S2 Text.**
(DOC)

## Acknowledgments

We thank the governments of Madhya Pradesh and Uttar Pradesh for their guidance and support provided during this study. We appreciate the doctors and community health workers who consented to participate and provided important information that helped us to arrive at the findings. We thank Dr. Manoj K Das, Director, Projects at INLEN Trust International, New Delhi for his guidance and support during this study. Special thanks to Dr. Rema Devi, Social Scientist, INCLEN for her valuable inputs to the manuscript.

## Author Contributions

**Conceptualization:** Rani Mohanraj, Shuba Kumar, Saradha Suresh.

**Data curation:** Rani Mohanraj, Shuba Kumar.

**Formal analysis:** Rani Mohanraj, Shuba Kumar.

**Methodology:** Saradha Suresh.

**Project administration:** Monica Agarwal, Bhavna Dhingra, Saradha Suresh.

**Resources:** Monica Agarwal, Bhavna Dhingra.

**Supervision:** Monica Agarwal, Bhavna Dhingra, Saradha Suresh.

**Writing – original draft:** Rani Mohanraj.

**Writing – review & editing:** Shuba Kumar, Monica Agarwal, Bhavna Dhingra, Saradha Suresh.

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
