## [Decision Letter · Decision Letter 0]

18 Jan 2022

PGPH-D-21-01136

Exploring challenges in the management of childhood pneumonia-Qualitative findings from health care providers from two high prevalence states in India

Dear Dr. Mohanraj,

Thank you for submitting your manuscript to PLOS Global Public Health. After careful consideration, we feel that it has merit but does not fully meet PLOS Global Public Health’s publication criteria as it currently stands. Therefore, we invite you to submit a revised version of the manuscript that addresses the points raised during the review process.

Additional Editor Comments (if provided):

Thank you for submitting this important manuscript to us. All three reviewers agree that this is a critical study and a well-written manuscript that would interest Plos GPH readers. They provide valuable recommendations and suggestions to help improve the manuscript to make it suitable for publication.

It is imperative to attend to reviewers’ comments. Some significant revisions include, but are not limited, to the following:

1. In the introduction and perhaps in the discussion, provide more context about the health system and its management structures for readers who may not be as familiar with the situation in the study area. This also would help make study findings and recommendations better tailored for implementation.

2. In the methods section, clarify the relationship between the larger study and the generation of this particular manuscript. Further, it would be useful to connect the theory or theories driving the study questions (e.g., health systems, community-based engagement) and how it/they connect(s) to the methods used. Use these theories to then in the interpretation of findings in the discussion. It might also be helpful to include some sample interview questions and codes, even as supplementary materials or as a table. Uploading the entire codebook is not necessary.

3. One reviewer stated that the results section as currently written is difficult to follow. You could revise to better reflect broader categories and sub-themes and integrate the quotes accordingly. The text should read more narratively, rather than a presentation of block quotes divorced from the summary. Additionally, it might be helpful to compare and contrast perspectives from different participant groups to understand the similarities and specific differences between them.

4. In the discussion section, reviewers have provided some helpful comments on how to structure this section such as beginning with a summary paragraph and then with a discussion of each of the key points in the following paragraphs. The manuscript can also be strengthened by linking study outcomes to specific interventions that have been successful in similar communities. Finally, it might be helpful to explore additional questions that remain unclear where further research is needed.

Journal Requirements:

We look forward to receiving your revised manuscript.

Kind regards,

Thurka Sangaramoorthy, Ph.D., M.P.H.

Section Editor

Reviewers' comments:

Reviewer's Responses to Questions

**Comments to the Author**

1. Does this manuscript meet PLOS Global Public Health’s publication criteria? Is the manuscript technically sound, and do the data support the conclusions? The manuscript must describe methodologically and ethically rigorous research with conclusions that are appropriately drawn based on the data presented.

Reviewer #1: Yes

Reviewer #2: Partly

Reviewer #3: Yes

2. Has the statistical analysis been performed appropriately and rigorously?

Reviewer #1: Yes

Reviewer #2: N/A

Reviewer #3: N/A

3. Have the authors made all data underlying the findings in their manuscript fully available (please refer to the Data Availability Statement at the start of the manuscript PDF file)?

Reviewer #1: Yes

Reviewer #2: No

Reviewer #3: Yes

4. Is the manuscript presented in an intelligible fashion and written in standard English?

Reviewer #1: Yes

Reviewer #2: Yes

Reviewer #3: No

5. Review Comments to the Author

Reviewer #1: The paper is very well written. There is so much rich content that might inform the improvement of management of pneumonia in similar settings.

Just a few comments:

1. The authors need to provide further detail about the organization of the health system. This is good for further contextualizing this phenomena and facilitates Knowledge translation. For instance, who is mandated to oversee healthcare provision in the states? Is it a decentralized system?

2. Related to the above point: an understanding of management structures of the health system could also help sharpen the recommendations. Which people in which offices should be held accountable or responsible for the findings. The recommendations(in the conclusion) need to be improved to mention which offices can be held accountable(or called upon to act)-it certainly can not be the entire government.

3. In the limitation, the authors mention that they needed a larger sample size. I think they are referring to more heterogeneity. The two are related but they could reconsider revising the term.

4. Under perceptions of CHWs(Page 10, line 234), they use the term "respiratory distress". This can be misleading, i suspect they want to mean a cough because that is what is managed with Cotrimoxizole. Kindly double check. If the term was misused, the maybe put it in inverted commas.

5. The authors may consider creating a table that summarizes the findings. As it is now, it may be difficult to navigate quickly.

Reviewer #2: This is a qualitative study regarding health care providers’ perceptions regarding management of childhood pneumonia in India. The data presented is a part of a larger mixed-methods study. The paper is well-organized and clearly written. The authors provide a good description of the study methodology and within the results section, data are presented within thematic categories which emerged through data analysis. The authors also discuss their data in relation to previous studies. There are some general take-away messages, but I think it might be more useful in the discussion section to link outcomes from the study (e.g., government hospitals access to supplies, training CHWs or other providers, community education and community trust in health care providers) to specific, feasible and potentially sustainable interventions that have been successful in similar communities. Some of this is in the discussion section (e.g., allowing CHWs to dispense antibiotics for pneumonia cases – though this approach could potentially lead to more unnecessary and inappropriate use of antibiotics in communities where they are already often misused). I would also be interested in seeing some information about what questions emerged or went unanswered during the study which might be important for follo-up research in the study sites, India, and/or region.

Reviewer #3: Thank you for the paper.

Title: add the after exploring

Equal contribution on authorship should be reserved for first and last author otherwise it seems overused

State the total number of participants in FGDs, again state the number per FGD and state how the composition.

Structure the Abstract appropriately

Seemingly some themes were deduced and not realised - that distinction should be clear e.g theme on challenges

Line 67-81- I find it confusing, it seems to be the previous study that led to the current one, if that is the case then it should be better summarised and show the linkage clearly

Methods

State the specific qualitative approach that was followed

Justify selection of the sites

Was there any theory that guided the study? If not, what informed the development of tools?

If you had no theory at point of design, acknowledge it as a limitation, however, use a theory to interpret your findings- you will achieve a better presentation and discussion of the results

Upload the guides that were used

Upload the Code book

How was coding done? deductive and inductive?

Results

Table 1- Group common variables like occupation- MDs, MOs. The current presentation of the table makes it difficult to follow as there are many repetitions that can be avoided if done better

For age- present the median with IQR if possible

Use either Barriers or Challenges but not both

The results need to be revised thoroughly, they are difficult to follow in the current state. You will need to have categories/subthemes under each theme and then show the quotes for each category presented. Analyse the system level results using the WHO building block framework. It will give the results better presentation and order

All the results would benefit from a better presentation and analysis

Employ constant comparison across the different groups so that one is able to appreciate areas of convergence and divergence- that will highlight the results better

there is an overlap between exclusive breastfeeding and nutrition - separating them makes them difficult to be viewed as such. revisit these aspects

Discussion

First paragraph should summarise the key findings in a paragraph and then discuss them from that summary.

I have pended review of the discussion till after the results are revised and presented in the suggested manner.

6. PLOS authors have the option to publish the peer review history of their article (what does this mean?). If published, this will include your full peer review and any attached files.

**Do you want your identity to be public for this peer review?** For information about this choice, including consent withdrawal, please see our Privacy Policy.

Reviewer #1: **Yes: **Phillip Wanduru

Reviewer #2: No

Reviewer #3: **Yes: **Alinane Linda Nyondo-Mipando, RNM, PhD

---

## [Decision Letter · Decision Letter 1]

7 Jul 2022

Exploring the challenges in the management of childhood pneumonia-Qualitative findings from health care providers from two high prevalence states in India

PGPH-D-21-01136R1

Dear Dr Mohanraj,

We are pleased to inform you that your manuscript 'Exploring the challenges in the management of childhood pneumonia-Qualitative findings from health care providers from two high prevalence states in India' has been provisionally accepted for publication in PLOS Global Public Health.

Best regards,

Meghnath Dhimal, Ph.D.

Academic Editor

Reviewer Comments (if any, and for reference):

Reviewer's Responses to Questions

**Comments to the Author**

1. If the authors have adequately addressed your comments raised in a previous round of review and you feel that this manuscript is now acceptable for publication, you may indicate that here to bypass the “Comments to the Author” section, enter your conflict of interest statement in the “Confidential to Editor” section, and submit your "Accept" recommendation.

Reviewer #4: (No Response)

2. Does this manuscript meet PLOS Global Public Health’s publication criteria? Is the manuscript technically sound, and do the data support the conclusions? The manuscript must describe methodologically and ethically rigorous research with conclusions that are appropriately drawn based on the data presented.

Reviewer #4: Yes

3. Has the statistical analysis been performed appropriately and rigorously?

Reviewer #4: N/A

4. Have the authors made all data underlying the findings in their manuscript fully available (please refer to the Data Availability Statement at the start of the manuscript PDF file)?

Reviewer #4: Yes

5. Is the manuscript presented in an intelligible fashion and written in standard English?

Reviewer #4: Yes

6. Review Comments to the Author

Reviewer #4: Well-written manuscript with description methodology and results

7. PLOS authors have the option to publish the peer review history of their article (what does this mean?). If published, this will include your full peer review and any attached files.

**Do you want your identity to be public for this peer review?** For information about this choice, including consent withdrawal, please see our Privacy Policy.

Reviewer #4: No
